# Gender Representation and Leadership in Local Transport Decision-Making Positions

**Lena Winslott Hiselius** [1,*] , **Annica Kronsell** [2] , **Lena Smidfelt Rosqvist** [3] , **Christian Dymén** [3] **and Olga Stepanova** [4]

1   Department of Technology and Society, Lund University, 22100 Lund, Sweden
2   School of Global Studies, University of Gothenburg, 40530 Gothenburg, Sweden; annica.kronsell@gu.se
3   Trivector Traffic, Vävaregatan 21, 22236 Lund, Sweden; lena.smidfelt@trivector.se (L.S.R.); christian.dymen@trivector.se (C.D.)
4   RISE Research Institutes of Sweden, 41258 Gothenburg, Sweden; olga.stepanova@ri.se
*   Correspondence: lena.hiselius@tft.lth.se

**Abstract:** This paper aims to analyse and further capture nuances of gender representation in local political decision-making bodies, focusing on implications for transport policy. Since gender is highly relevant for both attitudes towards transport policy as well as political votes, data on the gender and political colour of executives (members of presidiums) of transport-related committees, councils, and boards is analysed. The study is aimed at the local level, since municipal transport policy decisions include areas with clear differences between masculinity and femininity norms. The mapping of representation reveals, in line with other studies, that women are underrepresented in the most leading position (as chairperson of the City Board 31–37%), and that presidiums of transport-related committees, especially, are highly dominated by men (72–74%) with no clear positive trend in female representation identified over the studied years. The result suggests that transport-related decisions are disproportionally shaped by men as well as masculine norms, with implications for the transition towards transport sustainability.

**Keywords:** gender; sustainability; transport; Sweden; municipality; representation

## 1. Introduction

A long-standing concern in gender scholarship is how well women are represented in democratic institutions [1]. Research indicates that the unequal representation, as well as the absence of women in leading positions, may result in women's preferences being under-represented in agenda setting and in political decisions [2,3]. The role of gender in political leadership and decision-making has been analysed by scholars in several different fields. Gender-equal representation has been discussed in terms of its differential impact on policy, as substantive representation, and in relative terms as a critical mass, or as gender quotas [4,5]. Several concepts offer explanations for why women are unequally represented in political leadership, e.g., glass walls, leaking pipelines, and the glass ceiling [6]. Women tend to be underrepresented in higher positions of the political hierarchy and are less likely to be found in prestigious posts [7,8]. This is regardless of whether the person is an elected representative of a party [9,10]. Many scholars conclude, in line with Celis and Childs [1] (p. 28), that "political institutions privilege a masculinized political agenda and reproduce gendered norms".

This paper takes a comprehensive view of gender equality in the leadership functions in three local sites of political influence in Sweden: local committees taking decisions within the area of transport, the local City Board, and the City Council. This study aims to examine whether there is a discernible trend and disparity in female representation across various bodies, particularly in leadership positions, while accounting for political party affiliation. In this paper, we also ask whether women in leading positions increase the likelihood of

also having women in positions at lower levels, and especially whether women having the position of chairperson of the City Board influences the gender of the chairperson in other bodies.

The study was conducted for municipalities in Sweden and is based on a unique dataset collected in this study. Although statistics for the local level have previously been analysed for chairpersons in Sweden, e.g., Folke and Rickne [11] and Statistics Sweden [12], there are still few studies or datasets that capture the whole presidium, including the first and second vice chairperson. This study thus contributes to the research area by using a dataset with detailed information that enables representation per position analyses.

According to Homsy and Lambright [13], there is a growing body of research examining factors associated with gender in local government practices [14,15], but little is known about the influence of characteristics of the leaders of these bodies. Further, research has shown that factors such as the use of networking and affinity bias (the tendency for individuals to favour people who share similar characteristics or interests) contribute to the dominance of men in leading positions [16], and also at the local level of politics [17]. With an increasing share of female representation, it is also of interest to study whether the opposite applies, but so far there is a lack of these types of studies.

Much of the adjustment needed to reduce Sweden's emissions from transport must take place with the help of efforts at regional and local levels. Thus, Sweden's municipalities have an essential role. One of the municipality's most important planning tools is the land development plan (Översiktsplaner), as this plan provides guidance for how land and water areas should be used, preserved, and developed in the longer term in each municipality. This document may also be argued as giving the basic conditions for future transport infrastructure planning [18,19]. Within the area of urban planning and transport, municipalities take decisions on, e.g., parking strategies, the design of local streets, maintenance, and measures for sustainable travel.

As concluded in previous work [20], the transport area is, in the context of Sweden, comparatively unequal regarding the historic domination of men in decision-making and leadership. At the same time, it is important to reach set climate objectives and a sustainable and decarbonized transport system, and since it has been demonstrated widely that women show behaviours, preferences, and attitudes more in line with a climate agenda [21–25], female leadership in the transport sector is desirable.

The paper is structured as follows. It starts with a background section presenting research on gender inequalities, representation, and leadership in local politics in Sweden, which constitutes a base for the study outline. Thereafter, the results are presented and the paper ends with a discussion on the implications of the result for a more sustainable transport system.

## 2. Gender Inequalities, Representation, and Leadership in Local Politics in Sweden

This section presents the relevant literature and spells it out in terms of how gender equalities in representation and leadership can be explained, and presents what previous studies have shown about gender inequalities in the transport sector, for local politics, and particularly for Sweden. In doing so, it points to the research gaps that this study contributes to.

Several different concepts offer explanations for why men and women are unequally represented in political leadership, e.g., glass walls, leaking pipelines, and glass ceilings [6]. The term "glass walls" [26] is used to refer to invisible horizontal segregation 'walls' within an organization, where women end up in sectors or in functions, e.g., administrative roles, where the chances of career advancement are low. The term "the leaking pipeline" [27] denotes when more women leave their professional careers during the early stages and 'leak' from the career hierarchy, and thus there are fewer female than male candidates available when higher positions are to be filled. The term glass ceiling is used to describe how women are more disadvantaged than men the higher up they go in an organization, which makes it more difficult for women to reach the highest positions. Cotter et al. [28] suggest

that there is a glass ceiling when there is an unequal representation between the sexes that cannot be explained by other factors, that the inequality between the sexes increases with higher hierarchical levels, that there are circumstances that make advancement more difficult for women, and that inequality increases over time during the career. One factor put forward is that we tend to prefer people who are similar to ourselves. Within the area of politics, this has been shown to be a factor influencing women's representation, e.g., Crowder-Meyer [29] and Fox and Lawless [30]. Research by Niven [17] indicates that party chairs consistently preferred candidates more like themselves when analysing county party chairs and a sample of locally elected women from four states in the US.

Female political status is measured in various ways, as the percentage of women serving in legislative bodies or committees affiliated with those bodies, or as the percentage of women in administrative policymaking positions [20,31]. The concept of representation includes both the number and the political status/power of the representation [32]. Sweden is often described as a pioneering country when it comes to gender equality representation, and since the 1990s more than 40 per cent of the members of the parliament have been women and the governments have been largely equal numerically [33]. The country has implemented gender mainstreaming as a policy approach to actively promote and achieve gender equality [34], and has gained recognition for its notable performance on gender indexes, including the EU Gender Equality Index [35]. Since the 2014 Swedish elections, when all parties used some form of quota [5] there have been no formal quotas but, normally, an aspiration among parties to strive for equal representation. Studies show the effect of gender quotas and an increasing share of women in parliaments [5,36]. The notion of 'critical mass' suggests that a certain threshold of women's representation is necessary to bring about a significant impact on policies [37]. Dahlerup [4] proposed a minimum of 40% representation for influencing policy-making, but it is probable that this effect has diminished over time due to the increased prevalence of equality policies throughout the entire political spectrum. However, Bäck and Deus' [38] analysis of speechmaking among members of parliament in seven European countries (incl. Sweden) shows that female members of parliament take the floor less often when they are members of parties with high shares of female representatives, contradicting the critical mass theory.

A focus on only the share of female bodies obscures how power and influence relate to positions in an organizational hierarchy. The conditions for achieving and exercising leadership are not equal for men and women, and the gender inequality problems of political leadership remain relevant and women seem to have more difficulty in advancing to ministerial posts than men [39]. Sweden had to wait until 2021 to have its first female prime minister. Patterns of gender inequality also prevail at the local level. A mapping of representation in local committees that make decisions on transport-related issues [20,40] showed that women's representation in Swedish municipal committees has increased over the last 15 years, even though it is still far from equal. Compared with the City Council and City Board, the committees are much less equal. Men are overrepresented in municipal committees, such as on technical and building committees, and similar ones where transport planning and decisions are made [20]. This is in line with what other studies have shown on representation in the Scandinavian transport sectors [31], and also including the EU [41], in which transport issues are the domain of men.

Folke and Rickne [11] tested whether a glass ceiling existed for female municipal politicians elected from 1988 to 2010 in Sweden. Three levels were compared: ordinary elected members of the City Council, and chairpersons of the City Board and Council. The study shows that the underrepresentation of women in higher positions cannot be explained by the lack of female candidates or by the women who are at lower political levels lacking relevant qualifications. However, women in the sample made up 48 per cent of the elected members, 38 per cent of the chairpersons of committees, and 28 per cent of the chairperson of the City Board. The study shows that women are more disadvantaged the higher up they go in the hierarchy, and that the gender difference is greatest when it comes to the last political career step.

Other research suggests that political party affiliation may have relevance for gender-equal representation; for instance, parties with leftist ideologies favour equal representation and gender quotas and are more likely to appoint female leaders [42,43]. Figure 1, below, shows that this is the case for Sweden. The share of women in elected local assemblies is higher for left-wing parties (Social Democrat party, Leftist party, Green party). On the other hand, this was contradicted in Folke and Rickne's [11] analysis of locally elected persons after the elections of 2006 and 2010; they showed that the proportion of women decreased dramatically over that period, including in the leadership positions of the parties (both left- and right-wing parties).

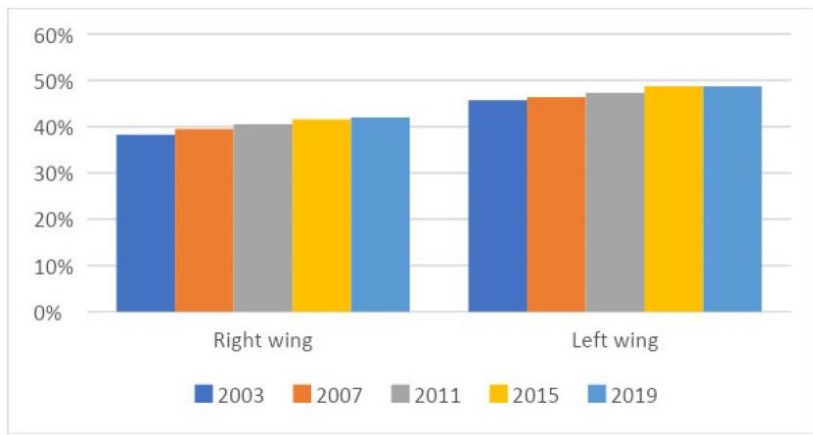

**Figure 1.** Share of female elected officials per party block (2003–2019). Figure based on data from Statistics Sweden [12].

The literature review indicates, altogether, a knowledge gap regarding representation in leading positions at the local level in Sweden. We will now continue by investigating a dataset consisting of individuals in presidiums of various bodies at the municipal level.

## 3. Local Government in Sweden: Method and Data

The material for this study was derived from local governments in Sweden. These local governments have far-reaching autonomy, which is enshrined in the constitution. The municipalities must comply with the framework set by the parliament and the government, but this autonomy gives each municipality the right to make independent decisions and, e.g., levy taxes on its inhabitants [18]. The City Council is the local parliament directly elected by citizens. The City Board is the local government, often based on a coalition between the parties with the most council seats. The chairperson of the City Board (considered the main leader of the municipality) is appointed by the largest party in the governing coalition after the general election, and the remaining board seats are distributed among all parties according to the number of seats they hold in the full assembly. Boards and Councils are responsible for the broader political decisions in the municipalities. The municipalities decide which committees they want, resulting in a varying organizational structure. The Council decides which committees should be present, and thus decide the organizational structure. A committee receives its mission from the Council and is responsible for a specific area, such as health, schools, buildings, and transport. They are responsible for the daily operations of the municipality, including preparing matters for the Council's decision-making and implementing Council decisions. The committees are empowered to make decisions on minor issues, as defined by the Council [18,19]. The leadership of the committees consists of a presidium, with a chairperson, a first vice and usually a second vice chairperson [12]. All positions within the City Board and the committees are appointed by the local parties by the rank order of the electoral ballot.

The analysis of political status connected to female representation was operationalized through a comparison of female representation (female share) in various positions in the presidiums of studied bodies. The analysis of power was carried out under the assump-

tion that different positions are connected to different political statuses. The position as chairperson is assumed to be connected to higher political status compared to the first vice chairperson, and even higher compared to the second vice. Further, it is assumed that the position as the chairperson of the City Board is assumed to be connected to higher political status than the chairperson of the City Council.

In this study, we also analysed differences in political status between men and women by studying the association between the gender of a person in a higher position and the gender of a person in a lower position (a vertical relationship). We focused on whether the association is significantly different depending on the gender of the person in the higher position.

Odds ratios were used to analyse the strength of the association between gender representation in different positions within and between bodies. The odds ratios were calculated as the ratio of the odds of an event occurring in one group to the odds of it occurring in another group, see, e.g., Persoskie and Ferrer [44]. Due to small sample sizes, Fisher's Exact Probability test was used to test whether the odds ratios were equal to 1 or not (i.e., no association). Finally, in order to analyse the trends in representation and political status, statistics for two years were compared.

The sampling frame of the survey consisted of all 290 municipalities in Sweden, and for this study unique data on representation and party affiliation for chairpersons (chairperson, first vice, and second vice chairperson) at committees, City Boards, and Councils for 2011 and 2015 were gathered and added from the municipalities through documents found on websites and contacts through email and telephone when documentation on websites could not be found. In municipalities in Sweden, there are often different committees in a municipality that take decisions related to the transport area. For these municipalities, the data of gender representation are not a binary number (man/woman) for a specific position in transport-related committees, but instead the share of men/women in the studied position.

For some municipalities, it was not possible to retrieve the information despite several attempts and reminders directed to local administrators. For representation in presidiums of transport-related committees, there was an initial limitation in the number (as identified in Winslott Hiselius et al. [20]) based on difficulties identifying the transport-related committees in the years 2011 and 2015. However, an analysis of the municipalities lacking in the dataset gave no indication of bias regarding geography, size, population, etc. The data collection process encompassed the years 2011 and 2015, allowing for the integration of this data with an existing dataset from a previous study. This merger resulted in a larger pool of information for analysis. In order to illustrate the size of the dataset in the following Tables, the total number of observations (total number of persons) is consistently presented along with the figures on the female share.

For the analysis, the bodies studied within each municipality were treated as distinct entities. Missing data for one body did not lead to the exclusion of data pertaining to other bodies in the analysis. This approach was chosen due to our focus on analysing different bodies rather than different municipalities. Only when examining the associations between bodies were the City Council and City Boards considered as a collective entity for each municipality.

## 4. Female Representation in Leading Positions

### 4.1. Share of Female in Presidiums

The first part of the analysis focused on the share of women in the presidiums of municipal bodies. The results are presented in Table 1. The figures show that the share of female representatives in the presidium was lower (and statistically significant so) in committees taking transport-related decisions than in the presidium of City Boards and Councils. This pattern was stable between the studied years, even though there was a positive trend over the studied years. The overall share increased from 2011 to 2015. However, only the average share for City Council presidiums (just) reached the threshold

for equality (40%) set by Dahlerup [4], Table 1. The low share of women in transport-related committees, only 26 and 28%, is in line with and supports previous research that identified the transport area as a male-dominated area, possibly influencing the agenda-setting to be masculine coded and possibly affecting the direction and content of decisions made, e.g., in less sustainable ways.

**Table 1.** Female share in presidiums (all positions) for years 2011 and 2015. In paratheses is the total number of persons in the analysis and the number of municipalities.

| Female Share in Presidiums of: | Year: 2011 | Year: 2015 |
| --- | --- | --- |
| City Board | 35% (507, 167) | 37% (544, 171) |
| City Council | 38% (800, 185) | 40% (810, 188) |
| Transport-Related Committee | 26% (673, 116) | 28% (705, 123) |

To analyse whether there was a difference in the share of women in presidiums of right-wing and left-wing party blocks, respectively, the data were separated based on the party affiliation of the persons in the presidiums. The result is presented in Table 2. The Moderate party, Christian Democrat party, Liberal party, and Center party are considered to be the right-wing party block, and the Social Democrat party, Leftist party, and Green party are considered to be the left-wing party block. At the local level, there are some minor parties represented, but they are usually not defined on the left–right scale and are also not relevant for this study, as the representatives rarely have a presidium position.

The results showed a generally higher share of women in presidiums representing left-wing party blocks, in line with the statistics in Figure 1. A slight increase in the gender share can be identified between 2011 and 2015 for both party blocks. The data reflect what we saw in Figure 1, where the share of women in elected local assemblies is higher for left-wing parties; this is also relevant for leadership positions, as seen in Table 2, below.

**Table 2.** Female share in presidiums (all positions) separated by party block for the years 2011 and 2015. In parentheses is the total number of persons in the analysis.

| Female Share in Presidiums of: | Year: 2011 | | Year: 2015 | |
| --- | --- | --- | --- | --- |
| | Right-Wing Party Block of the Presidium | Left-Wing Party Block of the Presidium | Right-Wing Party Block of the Presidium | Left-Wing Party Block of the Presidium |
| City Board | 31% (325) | 39% (311) | 34% (316) | 44% (326) |
| City Council | 32% (399) | 45% (360) | 34% (314) | 45% (378) |
| Transport-Related Committee | 23% (284) | 32% (185) | 23% (248) | 35% (249) |

*4.2. Share of Females for Different Positions in the Presidium*

Tables 1 and 2 hide a pattern of representation per position, as only the average is presented. In Tables 3 and 4, the statistics are separated according to position in the presidium. The low number of second vice chairpersons indicates that not all municipalities have appointed a second vice chairperson, especially for the transport-related committees studied.

The results in Table 3 indicate that, in line with previous research, the share of women in a high position (such as chairperson) was lower compared to the share in lower positions (such as first and second vice chairperson). This is also in line with previous research. For transport-related committees, the low share of women is more consistent for the positions studied. The biggest difference in gender share between positions within the same body could be found in City Councils. Further, there was no clear tendency for the gap in gender share between positions to diminish from 2011 to 2015.

**Table 3.** Female share per position for years 2011 and 2015 for studied bodies. In parentheses is the total number of persons in the analysis.

| Female Share | Year | |
|---|---|---|
| | **2011** | **2015** |
| **City Board** | | |
| Chairperson | 31% (242) | 37% (253) |
| First Vice Chairperson | 36% (224) | 38% (216) |
| Second Vice Chairperson | 39% (170) | 45% (173) |
| **City Council** | | |
| Chairperson | 29% (268) | 35% (265) |
| First Vice Chairperson | 43% (251) | 42% (245) |
| Second Vice Chairperson | 43% (240) | 41% (232) |
| **Transport-Related Committee** | | |
| Chairperson | 26% (193) | 29% (206) |
| First Vice Chairperson | 31% (183) | 30% (192) |
| Second Vice Chairperson | 19% (93) | 27% (99) |

**Table 4.** Female share per position and by party block for years 2011 and 2015 for studied bodies. In parentheses is the total number of persons in the analysis.

| Female Share | Year | | | |
|---|---|---|---|---|
| | **2011** | | **2015** | |
| | **Right-Wing Party Block of the Position** | **Left-Wing Party Block of the Position** | **Right-Wing Party Block of the Position** | **Left-Wing Party Block of the Position** |
| **City Board** | | | | |
| Chairperson | 26% (126) | 37% (116) | 31% (87) | 40% (166) |
| First Vice Chairperson | 32% (118) | 41% (106) | 32% (130) | 47% (86) |
| Second Vice Chairperson | 38% (81) | 40% (89) | 40% (99) | 51% (74) |
| **City Council** | | | | |
| Chairperson | 22% (139) | 36% (129) | 24% (105) | 43% (160) |
| First Vice Chairperson | 38% (138) | 50% (113) | 39% (127) | 46% (118) |
| Second Vice Chairperson | 37% (122) | 50% (118) | 36% (132) | 48% (100) |
| **Transport Related Committee** | | | | |
| Chairperson | 24% (108) | 28% (85) | 18% (99) | 40% (107) |
| First Vice Chairperson | 24% (90) | 38% (93) | 23% (93) | 36% (99) |
| Second Vice Chairperson | 21% (86) | 0% (7) | 30% (56) | 23% (43) |

To validate the representativeness, the figures were compared to statistics collected in other surveys. According to Statistics Sweden [12], the share of women as chairpersons of City Councils was 28% in 2010 and 36% in 2015, compared to 29% in 2011 and 35% in 2015 in this study (Table 3). In Folke and Rickne [11], the share of women as the chairperson of the City Board was 28%, based on data from 2006 and 2010 together.

Data separated for the party block of the position are presented in Table 4. Over the years studied, there was generally an increasing trend in the share of women in the studied positions, which could also be seen when separating the person holding the position by affiliation to either the right-wing or left-wing party block. The pattern was further consistent with the result in Table 2, where that the lowest share of women could be found in positions other than the chairperson, which could also be found when controlling for party block.

The results also reveal that the shares of women in the positions of the first and second vice chairperson were generally higher if the person in the position was affiliated with the left-wing party block than the right-wing party block (the large difference in the share of women as second vice chairperson between the right wing and the left wing was disregarded due to a limited number of observations).

Looking into the size of the difference in shares of women in lower positions compared to higher ones, there was a larger difference (a lower share of women as first and second vice chairperson compared to as chairperson) for the right-wing party block than for the left wing. This can be seen in City Boards and Councils. For the studied transport-related committees, there was no clear trend either over time or between positions. The committees did, however, show the lowest share of women among the studied bodies irrespective of party block.

The results indicate that women are overlooked or underestimated for leadership positions while being considered for supporting or deputy roles. Our results also illustrate the glass ceiling effect (the invisible barrier), which hinders women's upward career progression. Structural and cultural barriers seem to impede women's advancement to chairperson positions while allowing them to secure deputy chairperson roles. The same pattern is also revealed for the left-wing party block, despite a generally higher female share and this block having a more positive attitude towards equal representation and, e.g., gender quotas according to research [43].

### 4.3. Association between Positions at Municipal Level

In this section, we analyse the association between gender representation in different positions. Since there are municipalities where there is more than one committee involved in transport-related decisions (i.e., there is more than one chairperson and first and second vice person), this analysis was only carried out for City Boards and Councils.

#### 4.3.1. Association Gender Representation within a Municipal Body

We first analysed the association between gender representation in different positions within the same body using the odds ratio. We analysed whether the presence of a female chairperson raised the odds of having a female person in lower positions within the same body. Firstly, the number of cases was sorted by the gender of the chairperson of the City Board and the City Council and the gender of the first and second vice chairperson within the City Board and the City Council, respectively; see Table 5. The results in Table 6 present the odds and odds ratio and reveal that the presence of a female chairperson raised the odds significantly of having a male as the vice chairperson and lowered the odds of having a female person as the first vice chairperson. This was especially pronounced for the position of vice chairperson within the City Council for both 2011 and 2015. The pattern was less clear for the position of second vice chairperson.

#### 4.3.2. Association of Gender Representation between Bodies

In this section, we analyse a possible association between the gender of the highest position (chairperson of the City Board) and the gender of the chairperson and first and second vice chairperson on the City Council. As for the analysis of association within bodies, the number of cases was firstly sorted by the gender of the chairperson of the City Board and the gender of the first and second vice chairperson within the City Council; see Table 7. The results regarding odds and odds ratios are presented in Table 8. The odds

ratios presented were closer to one than when studying the association within the same body, indicating a lower influence of the gender of the chairperson of the City Board when analysing the association between bodies. The general pattern is that having a female chairperson on the City Board raises the odds of having a male person as chairperson at the City Council.

**Table 5.** Number of cases considering the gender of the chairperson and the gender of the first and second vice chairperson within the City Board and the City Council.

| | Year | | | | | | | |
|---|---|---|---|---|---|---|---|---|
| | **2011** | | | | **2015** | | | |
| **City Board Chairperson 2011, 2015** | **City Board First Vice Chairperson** | | **City Board Second Vice Chairperson** | | **City Board First Vice Chairperson** | | **City Board Second Vice Chairperson** | |
| | **Female** | **Male** | **Female** | **Male** | **Female** | **Male** | **Female** | **Male** |
| Female | 22 | 55 | 25 | 35 | 27 | 63 | 23 | 41 |
| Male | 63 | 101 | 47 | 71 | 69 | 87 | 60 | 68 |
| | **2011** | | | | **2015** | | | |
| **City Council Chairperson 2011, 2015** | **City Council First Vice Chairperson** | | **City Council Second Vice Chairperson** | | **City Council First Vice Chairperson** | | **City Council Second Vice Chairperson** | |
| | **Female** | **Male** | **Female** | **Male** | **Female** | **Male** | **Female** | **Male** |
| Female | 20 | 58 | 30 | 46 | 27 | 67 | 41 | 46 |
| Male | 93 | 98 | 78 | 97 | 87 | 92 | 65 | 100 |

**Table 6.** Odds and odds ratio of the gender of the chairperson and the gender of the first and second vice chairperson within the City Board and the City Council.

| | Year | | | | | | | |
|---|---|---|---|---|---|---|---|---|
| | **2011** | | | | **2015** | | | |
| **City Board Chairperson 2011, 2015** | **City Board First Vice Chairperson** | | **City Board Second Vice Chairperson** | | **City Board First Vice Chairperson** | | **City Board Second Vice Chairperson** | |
| | **Female** | **Male** | **Female** | **Male** | **Female** | **Male** | **Female** | **Male** |
| Odds when female | 0.40 | 2.50 | 0.71 | 1.40 | 0.43 | 2.33 | 0.56 | 1.78 |
| Odds when male | 0.62 | 1.60 | 0.66 | 1.51 | 0.79 | 1.26 | 0.88 | 1.13 |
| Odds ratio when female | 0.64 | 1.56 | 1.08 | 0.93 | 0.54 | 1.85 | 0.64 | 1.57 |
| *p*-value | 0.009 | | 0.011 | | 0.007 | | 0.011 | |
| | **2011** | | | | **2015** | | | |
| **City Council Chairperson 2011, 2015** | **City Council First Vice Chairperson** | | **City Council Second Vice Chairperson** | | **City Council First Vice Chairperson** | | **City Council Second Vice Chairperson** | |
| | **Female** | **Male** | **Female** | **Male** | **Female** | **Male** | **Female** | **Male** |
| Odds when female | 0.34 | 2.90 | 0.65 | 1.53 | 0.40 | 2.48 | 0.89 | 1.12 |
| Odds when male | 0.95 | 1.05 | 0.80 | 1.24 | 0.95 | 1.06 | 0.65 | 1.54 |
| Odds ratio when female | 0.36 | 2.75 | 0.81 | 1.23 | 0.43 | 2.35 | 1.37 | 0.73 |
| *p*-value | 0.009 | | 0.006 | | 0.006 | | 0.005 | |

**Table 7.** Number of cases considering the gender of the chairperson of the City Board and the gender of various positions in the presidium of the City Council.

| | Year | | | | | | | | | | | |
| | 2011 | | | | | | 2015 | | | | | |
| City Board Chairperson 2011, 2015 | City Council Chairperson | | City Council First Vice Chairperson | | City Council Second Vice Chairperson | | City Council Chairperson | | City Council First Vice Chairperson | | City Council Second Vice Chairperson | |
| | Female | Male | Female | Male | Female | Male | Female | Male | Female | Male | Female | Male |
| Female | 18 | 58 | 38 | 38 | 31 | 41 | 28 | 62 | 37 | 53 | 32 | 38 |
| Male | 51 | 113 | 62 | 102 | 62 | 91 | 57 | 99 | 62 | 94 | 65 | 87 |

**Table 8.** Odds and odds ratio of the gender of the chairperson of the City Board and the gender of various positions in the presidium of the City Council.

| | Year | | | | | | | | | | | |
| | 2011 | | | | | | 2015 | | | | | |
| City Board Chairperson 2011, 2015 | City Council Chairperson | | City Council First Vice Chairperson | | City Council Second Vice Chairperson | | City Council Chairperson | | City Council First Vice Chairperson | | City Council Second Vice Chairperson | |
| | Female | Male | Female | Male | Female | Male | Female | Male | Female | Male | Female | Male |
| Odds when female | 0.31 | 3.22 | 1.00 | 1.00 | 0.76 | 1.32 | 0.45 | 2.21 | 0.70 | 1.43 | 0.84 | 1.19 |
| Odds when male | 0.45 | 2.22 | 0.61 | 1.65 | 0.68 | 1.47 | 0.58 | 1.74 | 0.66 | 1.52 | 0.75 | 1.34 |
| Odds ratio when female | 0.69 | 1.45 | 1.65 | 0.61 | 1.11 | 0.90 | 0.78 | 1.27 | 1.06 | 0.94 | 1.13 | 0.89 |
| *p*-value | 0.011 | | 0.007 | | 0.007 | | 0.007 | | 0.005 | | 0.007 | |

## 5. Concluding Discussion

The results of this study indicate that when looking at the local leadership (the presidiums) of various bodies, there is a democratic problem regarding the representation of women in transport-related municipal decision-making, and that representation is unequal when analysing leadership roles. Through a unique dataset, we studied representation in leadership roles and found it to be unequal, with a generally low share of women in the presidiums. We also found a tendency to exclude women from leadership positions as chairperson (especially of City Boards), a position considered to be the most influential position of a municipality [12]. There was also a greater tendency to have a man as vice chairperson when a woman is a chairperson than the reverse. In this paper, we could thus identify various aspects of gender gaps at the local political level (either termed leaky pipeline or glass ceiling) when analysing different posts within presidiums, but also for the position as the chairperson of the City Board. When analysing a vertical relationship, i.e., if the gender of the person in the top post (chairperson of City Board) is associated with the gender of other posts in the presidiums, the result revealed a somewhat weaker association and indicated that aspects like affinity do not play out. This may also be a result of the way positions are appointed, based on a nomination by a party and not recruitment by the person with the most influential position. Our findings align with previous studies conducted on local representation in Sweden [20], as well as within the Scandinavian transport sectors [31], which have highlighted the prevalence of masculine dominance in the transport industry. Similarly, our study revealed a significant gender disparity, particularly in the highest political positions, corroborating the findings of Folke and Rickne [11]. Additionally, our results are consistent with research by Goodard [43] and

others, indicating that political parties with left-leaning ideologies tend to prioritize gender equality and are more inclined to appoint female leaders.

So, what are the implications of the result of this study on sustainable transport policies and measures mitigating climate change? The low representation of women in committees taking transport-related decisions supports the notion that the transport area is masculinized. The underrepresentation of women suggests that prevailing practices, planning, and structures prioritize other factors over the inclusion of women and the unique perspectives they offer [45]. Thus, as Celis and Childs [1] suggest, these are political institutions that privilege a masculinized political agenda and through it reproduce certain gender norms. Masculinized norms of technical solutions and infrastructure investments tend to overshadow alternative ideas [40]. This masculinized agenda may make it difficult for women to articulate diverging norms or alternative views. Also, in other sectors, representation is not enough to include a stronger gender equality perspective.

The discrepancy between men's and women's representation in leading positions that has been identified in this study through the gender share of the members of the bodies is close to equal and may be seen as a challenge since decisions on the local level are indicated to be important for sustainability [40]. Gender-equal representation may be relevant to increasing energy efficiency, as there are large discrepancies along the lines of gender in pro-environmental behaviour and travel patterns, as well as in attitudes and norms among citizens, planners, and decision-makers [25,46,47]. Women thus have behaviour, preferences, and attitudes more in line with a climate agenda, which makes their leadership desirable when climate change is to be tackled. The findings of these studies regarding gender differences in behaviour and attitudes indicate that if women were equally represented in policymaking and held equal influence in decision-making processes (including higher positions), the gender differentials observed would likely influence the outcomes of those decisions. Higher female representation in leading positions may also result in even more women in higher positions. As, e.g., critical mass theory suggested, there needs to be a certain volume to change norms and values.

The pressing time for policies mitigating climate change suggests that low gender equality is not "just" a democratic problem if we think that gender equality also could support more sustainable development. A question that might be asked is what representation measured as the percentage of women per body or presidium means? The results indicate that a focus on representation through the share of women out of total representatives in a body or of the presidium can simplify the question of gender equity and representation in local democratic institutions. There is a need to identify and follow up considering the degree of influence and power, as well.

**Author Contributions:** L.W.H.: Conceptualization, Methodology, Writing—Original draft preparation, A.K. and L.S.R.: Conceptualization, Methodology, Writing—Review and editing, C.D. and O.S.: Conceptualization, Methodology. All authors have read and agreed to the published version of the manuscript.

**Funding:** This work was supported by the Swedish Energy Agency [Grant Number 2019-021596].

**Institutional Review Board Statement:** Not applicable.

**Informed Consent Statement:** Not applicable.

**Data Availability Statement:** Not applicable.

**Conflicts of Interest:** No potential conflict of interest was reported by the author(s).

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
