# Peer review of "Gender Representation and Leadership in Local Transport Decision-Making Positions"

_sustainability, doi:10.3390/su151411280_

Round 1

Reviewer 1 Report

The manuscript entitled "Gender representation and leadership in local transport decision making positions", analyzes the influence of gender representation on transport sustainability in the near future. Actually, this work focused on the municipal transport policy decisions including areas with clear differences between masculinity and femininity criteria. It was found that a poor representation of women in the leading position will highly affect the sustainable programs toward an efficient transport system. Minor revisions are required for this manuscript as flows:

1.  The abstract must be including some numbers or percentages that reflect the important results that were found in the present study.

2.  If it is possible some tables or figures can be added to the introduction section to summarize the previous studies on the subject of gender representation in the transportation sector.

3.  More focus is needed to novelty of this work to be clearer for the readers.

4.  In the methods and data collection section on page 6, the authors sure work hard to collect a sampling frame of about 290 municipalities in Sweden, why the collected information is limited to the years 2011 and 2015 only.

5.  On page 8 line 287, please fix the reference problem "Tables Error! Reference source not found", to be clearer from the scientific approach.

6.  From a technique and political qualifications point of view, the results of Table 4 required more discussion to show the weakness of the share of females in the area of the City Board, City Council, and Transport-related committees.

7.  The comparison between the result of the present analysis and the results of other authors is needed.

8.  Please if it is possible add more modern references related to the subject for the years 2022 and 2023.

Author Response

  1. The abstract must be including some numbers or percentages that reflect the important results that were found in the present study.  

Thank you for your suggestion. The abstract has been revised accordingly. 

  1. If it is possible some tables or figures can be added to the introduction section to summarize the previous studies on the subject of gender representation in the transportation sector. 
  • Thank you for your good suggestion. However instead of adding tables or figures to illustrate the problem we have restructured the Introduction presenting the aim of the study earlier in the text compared to the previous version. This also makes the presentation of previous studies more coherent, and we believe illustration is not needed now. If however, you find it not sufficient please notify us and we will add such an illustration.  
  1. More focus is needed to novelty of this work to be clearer for the readers. 

- The text in the Introduction has been restructured in order to make the aim clearer. We have also added a sentence to the aim section in order to underline and clarify the contribution of our study.  

4.  In the methods and data collection section on page 6, the authors sure work hard to collect a sampling frame of about 290 municipalities in Sweden, why the collected information is limited to the years 2011 and 2015 only. 

  • This is now described in the Method section where we have included the following text section “The data collection process encompassed the years 2011 and 2015, allowing for the integration of this data with an existing dataset from a previous study. This merger resulted in a larger pool of information for analysis.”. Thus, this data collection was dependent on and built on previous data collection. Collecting data for more current time periods would of course be interesting, however it was considered too time-consuming for this study to start the data collection all over again.   
  1. On page 8 line 287, please fix the reference problem "Tables Error! Reference source not found", to be clearer from the scientific approach. 

- This problem seems to have been solved in the pdf version we received. We apologize if this problem was included in the original text we sent.   

6.  From a technique and political qualifications point of view, the results of Table 4 required more discussion to show the weakness of the share of females in the area of the City Board, City Council, and Transport-related committees. 

  • Thank you for pointing this out. An additional text section is not included elaborating on the result.   

7.  The comparison between the result of the present analysis and the results of other authors is needed. 

  • The text in the discussion has been revised accordingly now including comparisons with other studies mainly from the Swedish context.   

8.  Please if it is possible add more modern references related to the subject for the years 2022 and 2023. 

  • Thank you for pointing this out.  Five more references from 2021, 2022 and 2023 have been added in order to display the frontline of the research area.  
  • Ahrens, P.; Meier, P.; Rolandsen, A. The European Parliament and Gender Equality: An Analysis of Achievements Based on the Concept of Power. Journal of Common Market Studies. 2023, 61, 4, 1065-1081. doi:10.1111/jcms.13446  
  • Barnett, C.; Shalaby, M. All Politics Is Local: Studying Women’s Representation in Local Politics in Authoritarian Regimes. Politics & Gender. 2023, 1-6. doi:10.1017/S1743923X22000502 
  • Berevoescu, I.; Ballington, J. Women’s Representation in Local Government: A Global Analysis. 2021, New York: UN Women, Google Scholar 
  • Mechkova, V.; Dahlum, S.; Sanhuez Petrarca, C. Women's political representation, good governance and human development. Governance. 2022, 1–20, doi:10.1111/gove.12742 
  • Kim, N. When does women’s political power matter? Women’s representation and legal gender equality of economic opportunity across contexts. European Political Science Review. 2022, 14, 4, 583-599. doi:10.1017/S1755773922000352 

Reviewer 2 Report

1.  Research Motivations and Research Questions: The introduction section could be structured better. The main research questions of the study were buried by that materials on page 2 (line 42-58), which had multiple threads and story lines. I understand that you were trying to identify the gaps in literature and articulate your contributions. But I think it turned out to be ineffective and distracting – the info here is more about the data (how you address the research questions) instead of the story itself. It might be helpful to move the research-question paragraph (line 60 – 68) up as the second paragraph on page 2. Line 42-58 could be a part of the contributions of the paper (which can come after laying out the key story/research questions).

 2.       The central theme of the story: the paper seemed to lack a well-articulated structure or logic to address the research questions of “the trend and differences in female representation” (line 63; BTW, this sentence is not grammatically correct).

 -          You talked about two issues in female representation: presence (in number) and power/influence aspects of gender status. How would your data and reporting tables align with these two distinct components?

-          You did not clearly define or describe the methodology you used to operationalize the “political status of female representation” and used multiple terminology throughout the paper such as number, or share, or magnitude, N, % etc.

-          You did not provide sufficient information to help readers understand what each number and symbol in your tables meant and not every table used the same format (e.g., Table 4 was different from Table 1 – 3).  

-          You did not provide information on how you handled the missing data: should the three levels (board, council, and committee) be treated as set and any incomplete set should be removed from the data for the final analysis? For instance, if the information for the transport committee was not available for a particular city, should the board and council data for the city be included or removed?

Extensive editing needed to address the grammar errors, in-precision language issues, and typos.

Author Response

1. Research Motivations and Research Questions: The introduction section could be structured better. The main research questions of the study were buried by that materials on page 2 (line 42-58), which had multiple threads and story lines. I understand that you were trying to identify the gaps in literature and articulate your contributions. But I think it turned out to be ineffective and distracting – the info here is more about the data (how you address the research questions) instead of the story itself. It might be helpful to move the research-question paragraph (line 60 – 68) up as the second paragraph on page 2. Line 42-58 could be a part of the contributions of the paper (which can come after laying out the key story/research questions). 

  • Thank you for pointing this out. We have restructured the Introduction based on your suggestion and the aim is now presented earlier. We hope that this gives the text a clearer red thread.  

2. The central theme of the story: the paper seemed to lack a well-articulated structure or logic to address the research questions of “the trend and differences in female representation” (line 63; BTW, this sentence is not grammatically correct). 

  • Thank you for pointing out the incorrect sentence. This is now hopefully correct. 
  • Regarding the structure of the analysis, we have added a text section in Method that describes the logic behind our analysis and how it correlates with the research questions. This section also gives a structure for the results presented. We hope this text section clarifies our thoughts and intentions. Please notify us if there are still uncertainties.  

-          You talked about two issues in female representation: presence (in number) and power/influence aspects of gender status. How would your data and reporting tables align with these two distinct components? 

  • Thank you for suggesting this clarification - indeed this was missing previously. We interpret this question as relating to the previous one. We have improved the text in the Metod section in order to better describe the method and how our analyses relate to the research questions asked. 

-          You did not clearly define or describe the methodology you used to operationalize the “political status of female representation” and used multiple terminology throughout the paper such as number, or share, or magnitude, N, % etc. 

  • Thank you for suggesting this clarification - indeed this was missing previously. We interpret also this question as relating to the previous one. We have improved the text in the Metod section in order to better describe the method and how our analyses relate to the research questions asked. 
  • And thank you for pointing out the multiple terminology. Describing text relating to the tables and in the tables has been revised in order to ensure a coherent presentation of the variables calculated. We have also split up Table 5 and 6 in order to handle only one type of data per Table. Hopefully, this makes the tables easier to comprehend.   

-          You did not provide sufficient information to help readers understand what each number and symbol in your tables meant and not every table used the same format (e.g., Table 4 was different from Table 1 – 3).   

  • Thank you for suggesting this clarification. We interpret this question as relating to the previous one. Describing text relating to the tables and in the tables has been revised in order to ensure a coherent presentation of the variables calculated.  

-          You did not provide information on how you handled the missing data: should the three levels (board, council, and committee) be treated as set and any incomplete set should be removed from the data for the final analysis? For instance, if the information for the transport committee was not available for a particular city, should the board and council data for the city be included or removed? 

  • Thank you for suggesting adding information on missing data. We have added a text section in the Method section describing this problem and how we have handled it. ” For the analysis, the bodies studied within each municipality were treated as distinct entities. Missing data for one body did not lead to the exclusion of data about other bodies in the analysis. This approach was chosen due to our focus on analyzing different bodies rather than different municipalities. Only when examining the associations between bodies, the City Council and City Boards were considered as a collective entity for each municipality.”  

Round 2

Reviewer 2 Report

I truly appreciate authors' efforts to revise the manuscript. The quality of the story is much improved. 

Minor correction and polish would be appreciated.